# Surface Modification of Dielectric Substrates by Broad Beams of High-Energy Atoms of Inert Gases

**Alexander Metel \***, **Sergey Grigoriev, Marina Volosova, Yury Melnik** **and Enver Mustafaev**

Department of High-efficiency Processing technology, Moscow State University of Technology,
Vadkovsky per. 3A, 127055 Moscow, Russia; s.grigoriev@stankin.ru (S.G.); m.volosova@stankin.ru (M.V.);
yu.melnik@stankin.ru (Y.M.); e.mustafaev@stankin.ru (E.M.)
* Correspondence: a.metel@stankin.ru; Tel.: +7-903-246-4322

**Abstract:** We present a new method to generate a neutral beam for surface treatment of materials by fast atoms of inert gases. The new method allows for treatment at lower pressures enlarging the scope for glow discharge applications. To generate the monoenergetic neutral beam, a grid composed of parallel plates is placed inside a vacuum chamber, a glow discharge plasma was generated, and a beam was formed by pulsing the grid to 30 kV to extract ions from the glow discharge. The ions were then neutralized by small-angle scattering at the surfaces of the grid. By applying the high voltage for 50 µs with a repetition frequency of 50 Hz, heating of the target could be limited to 100 °C (instead of 700 °C when running continuously). We present results showing the uniformity of the created beam and its energy distribution using Doppler-shift measurement. Finally, we show friction measurement of treated alumina pieces as a working example of an application of this technology.

**Keywords:** glow discharge; electron confinement; plasma homogeneity; charge-transfer collisions; accelerated ions; fast atoms; implantation

---

## 1. Introduction

Ion implantation is a method for injecting high-energy atoms into the surface of a solid to modify its near-surface layer [1–5]. There can be modifications to chemical, physical, mechanical properties and microstructure of the layer. They can appear as changes in hardness, wear-resistance, stiffness, friction response, corrosion behavior or other mechanical properties such as fatigue and contact fracture toughness [6–11].

The implantation is carried out in vacuum with ion beam energies of ~100 keV and ion current densities typically between a few microamps to one milliamp per square centimeter. To generate the ion beams, accelerators of charged particles have been developed. They are characterized by a comparatively high cost and long processing time [12–14]. Moreover, the beam cross-section of those implanters is limited. To process large substrates or three-dimensional products, it is necessary to move and rotate them in a vacuum.

To modify the surface of technical parts, plasma immersion ion implantation is widely used [15]. Usually, the products are immersed in a gas discharge plasma, which contains ions to be implanted. To accelerate the ions, negative high-voltage pulses with amplitude up to 100 kV are applied to the products at the gas pressure not exceeding 0.1 Pa [16]. At higher pressures, the energy of ions bombarding the product diminishes. This is due to charge exchange collisions [17,18] in the space charge sheath [19] between the product and the plasma. The advantage of this method is that in each pulse, the whole product surface is processed and there is no need to move it.

It is impossible to apply the pulses to dielectric products. For this reason, broad beam sources of ions [20–24] or fast neutral atoms [25,26] are used to treat them. To produce a beam of fast atoms,

it is necessary to first generate at a pressure of 0.1–1 Pa a large homogeneous plasma emitter of ions, to extract ions from the emitter and to accelerate them using a multigrid system and to neutralize the space charge of the ion beam by electrons. Then it is necessary to convert the ions into fast neutral atoms due to charge exchange collisions with gas molecules and separate them from residual ions using a magnetic field.

On the base of the hollow cathode glow discharge, broad beam sources of fast molecules have been developed [27,28]. They are remarkable for simple design, one-grid ion-accelerating system and ability to produce fast atom beams of reactive gases combined in some cases with flows of metal atoms. It is quite tricky to produce a reactive plasma emitter of ions at a high positive potential. Therefore, the highest energy of atoms produced by those sources amounts to 10 keV. The main difficulty is high-voltage insulation of the source anode and hollow cathode. It would be much easier to apply the negative high voltage to an emissive grid immersed in a plasma, whose potential difference with the grounded chamber does not exceed one kilovolt. It was shown in [29] that in the latter case, broad beams of fast neutral atoms could be produced.

The goal of the present research is to develop a technology, which could increase the rate of modifying the near-surface layers of dielectric substrates bombarded by high-energy neutral atoms and allow processing of the substrates with larger surface areas.

## 2. Experimental

Figure 1 presents an experimental setup with a 50-cm-long and 50-cm-diameter process chamber with two doors. On the chamber top, there was feedthrough. It allowed the connection of any product placed inside the chamber to power supplies. On the left door of the chamber an $18 \times 16 \times 12$ cm rectangular housing was mounted with a removable flange cooled with water. It allowed fastening on its inner surface substrates to be processed. Inside the housing, a grid was fixed on ceramic insulators (not shown in Figure 1). It was composed of 21 titanium $140 \times 50$-mm plates 0.5-mm-thick and 7.5 mm apart.

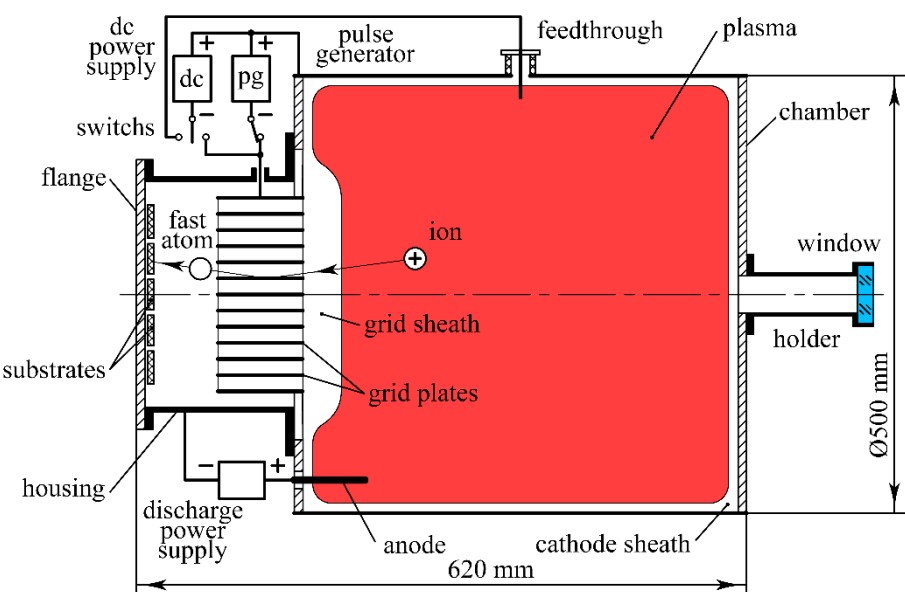

**Figure 1.** Schematic of an experimental setup for modification of dielectric substrates.

The housing was connected electrically to the grounded chamber. The negative pole of a discharge power supply, which ensures a stabilized current up to 4 A, was connected to the chamber. Its positive pole was connected to the anode placed inside the chamber. A generator of high-voltage pulsed could apply between the chamber and the grid 50-μs-wide pulses with amplitude up to 30 kV and a repetition frequency of 50 Hz. The pulse generator could be replaced with a DC power supply. The latter enabled

the application of stabilized negative voltage up to 5 kV to the grid or a product inside the chamber fastened to the feedthrough.

The chamber volume of 0.12 m$^3$ was evacuated to ~0.001 Pa by a turbomolecular pump. In the process chamber, the operating pressure of argon was maintained by a gas control system and was measured with an ion gauge and a BARATRON gauge. There was a quartz window on the right door of the chamber. It allows for remote measurements of the substrate temperature with an infrared pyrometer IMPAC IP 140. Through the window, the image of the grid was focused using a quartz lens with a 75-mm focal-length onto the entrance slit of an STE-1 spectrograph (not shown in Figure 1).

## 3. Results

In the first experiments, the multiplate grid was dismounted from the housing and hanged upon the feedthrough in the center of the process chamber. The 250-mm-diameter opening between the chamber and the housing was covered with a steel disc.

After the chamber was pumped to a pressure of 1 mPa, argon was supplied to it. At the gas pressure $p \sim 0.5$ Pa switching on the discharge power supply results in establishing the hollow cathode glow discharge with voltage amounting to ~400 V. The chamber played the role of the hollow cathode with an inner surface area of $S = 1.5$ m$^2$ and volume of $V = 0.12$ m$^3$. It was filled with a bright and quite homogeneous glow of plasma separated from the chamber walls with a cathode sheath.

When the DC power supply connected to the grid was switched off, the grid and chamber were equipotential. In this case, the cathode sheath width of several mm was approximately equal to the sheaths near the grid plates. At a constant discharge current $I_d$ in the anode circuit, the discharge voltage $U_d$ between the anode and the chamber did not practically change with argon pressure decreasing down to $p \sim 0.1$ Pa. However, with further pressure decreasing, it rose to $U_d \sim 1$ kV at $p = 0.02$ Pa. When at a constant current $I_d = 2$ A the voltage $U$ between the chamber and the grid rose from zero up to 5 kV, then the width $d$ of the sheath between the plasma and the grid rose to ~3 cm. The current $I$ in the grid circuit grew by several times due to the current of secondary electrons emitted from the grid, which exceeded the ion current in the grid circuit. Bombardment of the chamber walls by those electrons increased the electron emission from the chamber. As a result, the discharge voltage $U_d$ diminished to ~200 V.

After running the discharge at $p = 0.1$ Pa, $U = 5$ kV and $I = 1$ A for 20 min, two etched prints were found on both chamber doors. One of them was observed on the steel disc, which covers the housing opening (Figure 2). The other was etched on the right door close to the holder of the quartz window.

At the pressure of 0.1 Pa, the density $n$ of gas atoms amounted to $2.5 \times 10^{19}$ m$^{-3}$. For 5-keV argon ions the cross-section of charge transfer collisions was equal to $\sigma = 1.5 \times 10^{-19}$ m$^2$ [17,18]. The charge-transfer length for these ions was equal to $\lambda = 1/(n\sigma) \sim 27$ cm, which by an order of magnitude exceeded the sheath width $d$. Thus, the fast atoms could not appear due to charge-transfer collisions of argon ions in the space charge sheath surrounding the grid as it occurred in [27] at $p \sim 1$ Pa. Accelerated in the sheath argon ions enter the gaps between the grid plated. Due to the angular divergence of the ions caused by inhomogeneity of the electrical field near the edges of the plates, most of them collided with the plates at a high angle of incidence exceeding 85 degrees. In these conditions, ions did not sputter the plates. They pulled out from the metallic surface electrons, which neutralized their charges. As a result, ions turned into fast neutral atoms, which were reflected off the plated and continued their way out of the grid.

The print on a steel disc (Figure 2) was etched mainly by fast atoms produced in the central part of the grid. Ions entering peripheral gaps of the grid collided the plates at lower angles of incidence. They sputtered at one of the plate sides and, hence, did not turn into fast atoms. Thus, though the quasirectangular print was similar to the grid cross-section, the print height of ~12 cm was less than the grid height of 14 cm. The print width of ~14 cm was less than the grid width of 16 cm. The outer surfaces of the left and right plates were sputtered by ions accelerated from the plasma surrounding the grid the most intensively. At a voltage of $U = 5$ kV applied to the grid and current in its circuit

of $I = 1$ A, a substantial part of electrical power $U \times I = 5$ kW was spent on sputtering and heating the outermost plates and acceleration of secondary electrons emitted by their surfaces. Due to the sputtering of the grid, the chamber walls were coated with titanium films.

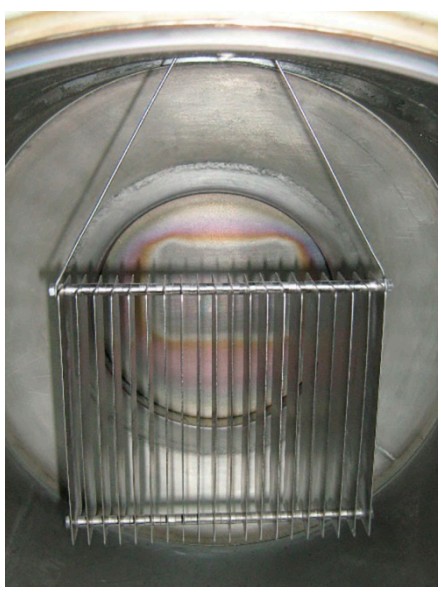

**Figure 2.** Inner view of the process chamber after etching a steel disc by 5-keV argon atoms produced by the grid composed of parallel plates.

To eliminate the above drawbacks, the multiplate grid was returned to its place in the housing (Figure 1). Here the grid could be connected to the same DC power supply or generator of high-voltage pulses. Edges of the grid plates lied in the plane of the inner wall of the chamber door. The area of the grid side facing the chamber amounted to $S_g \sim 0.024$ m$^2$.

At the discharge current of $I_d = 2$ A, current in the grid circuit was equal to $I_g = 0.26$ A. At the ion current density $j_i = I_g/S_g \sim 10$ A/m$^2$, the width of the cathode sheath near the entrance to the housing amounted to ~3 mm. This was less than half the distance between the grid plates amounting to 3.75 mm. However, the high width of the grid plates amounting to 50 mm prevented the plasma penetration from the chamber into the housing.

When a bias voltage of 3 kV was applied to the grid, the width of the sheath between the plasma and the edges of grid plates grew to ~13 mm. In any case, the surface of plasma-emitting ions towards the grid was flat. It allows the formation of a well-collimated broad beam of ions entering the grid. The grid transparency η could be assessed as the ratio of the 7.5-mm distance between the grid plates by the sum of the distance and the plate thickness of 0.5 mm. It yielded the transparency of η = 0.94. Hence, only 6% of accelerated ions sputter the edges of the grid plates. The rest entered the gaps between them—and after contacts with the plate surfaces—turned into fast neutral atoms.

To assess the homogeneity of the fast atom beam, five 2-mm-thick, 30-mm-high and 48-mm-wide substrates were fastened on the housing flange (Figure 1). They were made of pure polycrystalline aluminum oxide $\alpha$-Al$_2$O$_3$. One side of each substrate was polished and covered with a 1-mm-thick, 10-mm-wide and 48-mm-long titanium mask. During the bombardment of the substrate by the beam, the temperature of the mask placed on the intermediate substrate was measured by pyrometer IMPAC IP 140. At the discharge current of $I_d = 2$ A and the voltage between the chamber and the grid $U = 3$ kV, the mask temperature in ten minutes grew to ~700 °C.

After 2-h-long processing, the substrates were taken off the chamber. The masks were detached from the substrates, and using stylus profiler Dectak XT produced by Bruker Nano, Inc. (USA), the height δ of the steps between covered by the masks and open surfaces of the substrates was

measured (Figure 3). For the intermediate substrate, it amounts to δ = 1.2 μm, thus indicating that the sputtering rate was equal to v = 0.6 μm/h.

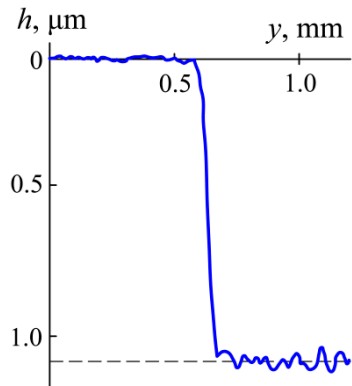

**Figure 3.** Profilogram of an $Al_2O_3$ substrate surface bombarded with a mask by fast argon atoms.

Due to homogeneity of the discharge plasma near the grid, the width of the space charge sheath between the grid and the plasma was distributed quite uniformly. For this reason, the beam of fast argon atoms bombarding the substrates was well collimated. This was confirmed by an abrupt transition between the substrate surfaces in Figure 3 covered with the mask (on the left) and open (on the right), as well as a homogeneous distribution of the substrate etching rate (Figure 4).

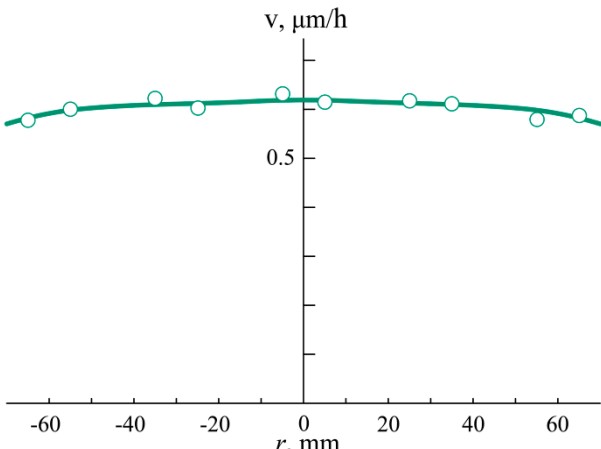

**Figure 4.** Substrate etching rate v versus the distance *r* from the center of the flange.

Dependence of the fast atom energy $\varepsilon$ on the voltage $U$ applied to the grid was studied using spectrograph STE-1. Figure 5 presents two spectrograms of helium discharge plasma at the discharge current of $I_d$ = 2 A and voltage between the chamber and the grid $U_1$ = 3 kV and $U_2$ = 1 kV. Spectral lines emitted by ions or atoms moving towards the spectrograph or in the opposite direction were due to the Doppler effect shifted to the short-wave region of the spectrum or its long-wave region. The Doppler shift Δλ was proportional to the velocity $v$ of the particles

$$\Delta\lambda = \lambda v/c, \tag{1}$$

where $c$ is the velocity of light and λ is the wavelength of the line. At equal kinetic energies $\varepsilon$ velocity

$$v = (2\varepsilon/M)^{1/2} \tag{2}$$

of helium atoms, where $M$ was their mass, more than three times exceeds the velocity of argon atoms, whose mass was by ten times higher. Accordingly, the Doppler shift for helium more than three times

exceeds the shift for argon. Hence, the velocity of fast helium atoms could be measured with higher precision compared to argon. Therefore, for investigations of the dependence on the voltage $U$ of fast atoms energy, helium was chosen.

Three helium lines are presented on the spectrograms of Figure 5; their wavelengths are equal to $\lambda = 386.763$ nm, $\lambda = 387.179$ nm and $\lambda = 388.865$ nm. All of them were emitted by slow gas atoms excited in the plasma filling the chamber. At $U = 3$ kV, on the left side of the line with $\lambda = 388.865$ nm, a satellite was shifted from the line center at $\Delta\lambda_1 = 0.52$ nm. The satellite was emitted by fast atoms moving from the grid towards the substrates, i.e., in the direction opposite to that of photons moving through the quartz window to the entrance slit of the spectrograph.

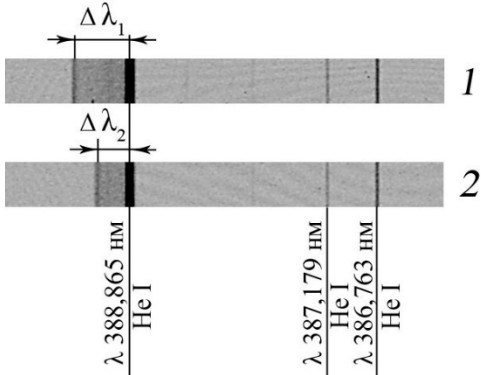

**Figure 5.** Spectrograms of the discharge plasma at helium pressure $p = 0.3$ Pa and voltage between the chamber and the grids $U = 3$ kV (1) and 1 kV (2).

As the density of slow gas atoms exceeded the density of fast atoms by several orders of magnitude, the intensity of the line with $\lambda = 388.865$ nm significantly exceeded the satellite intensity. Thus, the line with $\lambda = 388.865$ nm was overexposed and its "width" on the photographic film significantly exceeded the width of the lines with $\lambda = 387.179$ nm and $\lambda = 386.763$ nm. Satellite intensities of the following lines were so weak that they were not registered on the film.

The kinetic energy of fast atoms could be calculated using formula

$$\varepsilon = (Mc^2/2)(\Delta\lambda/\lambda)^2 \tag{3}$$

For $\Delta\lambda_1 = 0.52$ nm, $\lambda = 388.865$ nm, velocity of light c = $3 \times 10^8$ m/s and the helium atom mass $M = 6.64 \times 10^{-27}$ kg the Formula (3) yields the energy of helium atoms equal to $\varepsilon = 5.36 \times 10^{-16}$ J = 3300 eV. This was in good agreement with the value of $e(U + U_d)$, where $e$ was the electron charge and $U_d \sim 300$ V was the discharge voltage between the anode and the chamber. At $U = 1$ kV, the satellite was closer to the line center. Its shift was equal to $\Delta\lambda_2 = 0.31$ nm and yields the energy of fast atoms $\varepsilon \sim 1300$ eV.

It could be supposed that fast atoms produced due to the charge neutralization of accelerated by the grid ions on the surfaces of the grid plate were practically monoenergetic. Their energy was defined by the potential difference between the plasma and the grid. It was equal to the sum of applied to the grid accelerating voltage $U$ and discharge voltage $U_d$ between the anode and the chamber. When $U$ grows to 10–30 kV, the discharge voltage may be neglected. Then the fast atom energy was supposed to be approximately equal to $eU$.

Application to the grid of 50-µs-wide 30-kV pulse resulted in irradiation of the substrates with fast helium or argon atoms. At the discharge current $I_d = 2$ A, the growth of the voltage applied to the grid did not influence the ion current density $j_i = 10$ A/m$^2$ in the grid sheath. However, the sheath width increased to 70 mm.

At the repetition frequency of the pulses amounting to 50 Hz, the mean density of the heating power transported to the substrates by the fast atoms was equal to $3 \times 10^4 \times 10 \times 5 \times 10^{-5} \times 50 =$

750 W/m$^2$. This was by 40 times less than the heating power density at the continuous application to the grid of $U = 3$ kV. For this reason, during the substrates processing by pulsed beams of fast atoms, their temperature only slightly exceeds 100 °C.

When all accelerated ions after passing through the grid turn into fast neutral atoms and were implanted into the substrate surface, the beam current density of $j_i = 10$ A/m$^2$ corresponds to the implantation rate of $j_i/e = 10/1.6 \times 10^{-19}$ m$^{-2}$ s$^{-1} = 6.25 \times 10^{15}$ cm$^{-2}$ s$^{-1}$. Hence, it would take only 16 s to obtain the implantation dose of $10^{17}$ cm$^{-2}$. Taking into account a possible absorption of accelerated ions by the grid plated and reflection of fast atoms from the substrate surface, the implantation rate could be lower by several times. For this reason, in our experiments, the irradiation time was equal to 30, 60 and 90 s.

To study the influence of irradiation with fast helium and argon atoms on friction coefficient of the substrate surface was used as a test machine, Tetra Basalt N2 precision tribometer (Falex Tribology NV, Rotselaar, Belgium). A 4-mm ball made of SiC was used as the counter body. The tests of all substrates were carried out under conditions of dry friction at identical normal loads on the counter body (1.0 N), with the rotation speed of 3 RPS at the trajectory radius of 5 mm, i.e., the relative displacement speed of 9.4 cm/s and sliding distance of 1000 m (1600 cycles).

The evolution of the friction coefficient during the tests was presented in Figure 6 for an untreated substrate and substrates irradiated with 30-keV argon atoms for 30, 60 and 90 s.

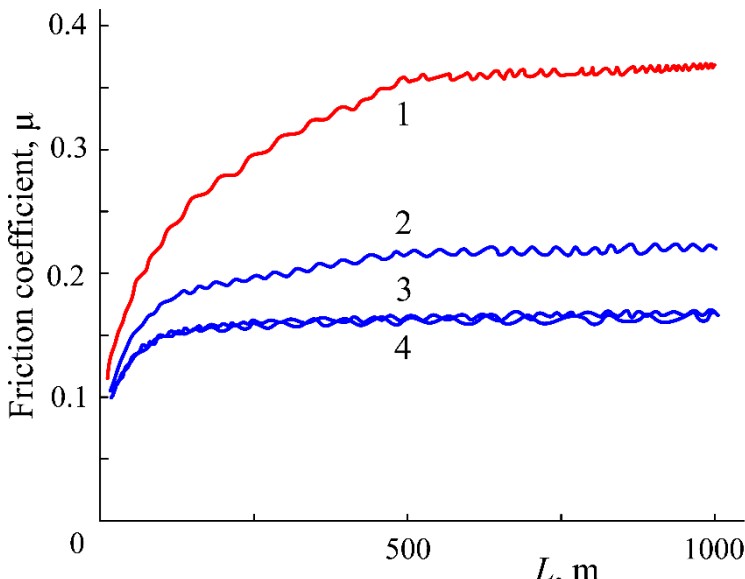

**Figure 6.** Dependence of the friction coefficient μ on the sliding distance $L$ for an untreated substrate (curve 1) and substrates irradiated with 30-keV argon atoms for 30 (curve 2), 60 (curve 3) and 90 s (curve 4).

In all cases, the initial value of the friction coefficient was relatively low. However, during the test, it grows and reaches a stable value of 0.37 for the untreated substrate and of 0.21, 0.16 and 0.15 for the substrates irradiated for 30, 60 and 90 s. It means that the substrate irradiation appreciably decreased the friction coefficient.

Low values of the friction coefficients during the early phase of friction tests could be explained by adsorption to the substrate surface of the water, which acted as a lubricant and disappears during the test. The decrease in the friction coefficient of the irradiated substrate could be attributed to the modification of its surface.

## 4. Discussion

Application to conductive products immersed in plasma of high-voltage pulses at the gas pressure of ~0.01 Pa enables the plasma immersion ion implantation for various products [30–39]. At this pressure, ions accelerated in the sheath between a conductive product and plasma pass through the sheath without collisions. Hence, their energy exactly corresponds to the amplitude of applied pulses and amounts to 10–100 keV. This allows the surface hardening of massive steel products immersed in plasma [40] and other technological processes. It was shown in [41] that electrostatic confinement of electrons inside a sizeable hollow cathode reduces the gas pressure in the glow discharge to 0.01 Pa. This finding substantially enlarged the scope of the glow discharge applications. Any process vacuum chamber can play the role of the hollow cathode and can be filled with a homogeneous discharge plasma [42].

For bombardment by high-energy particles of dielectric products, it was proposed to use a high-transparency grid immersed in plasma (Figure 7). When a negative high voltage is applied to the grid, it is surrounded by a space–charge sheath. Accelerated from the plasma ions pass through the grid holes, are decelerated on the other side of the grid and change direction of their movement.

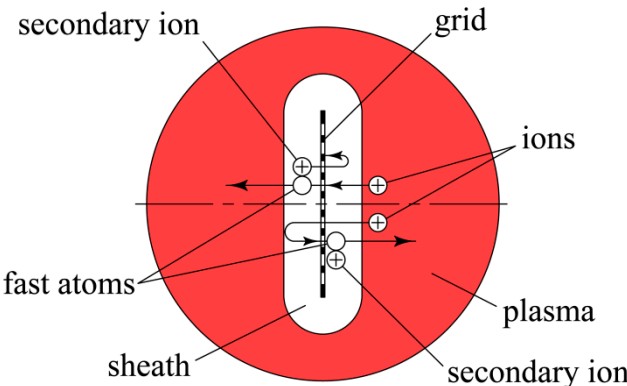

**Figure 7.** Schematic of fast atoms production in the sheath of a flat grid immersed in plasma.

The mean way of ions reciprocating through the grid amounts to

$$L = 2d/(1 - \eta), \qquad (4)$$

where $d$ is the sheath width and $\eta$ is the grid transparency. At the ion current density of $j_i = 10$ A/m$^2$ and applied to the grid bias voltage of three kilovolts, the width of the sheath between the plasma and the grid amounts to $d = 1.3$ cm. When the grid transparency $\eta$ is equal to 0.8, the length of the ions way in the sheath amounts to $L = 13$ cm. At argon pressure $p = 0.1$ Pa the charge-transfer length for these ions is equal to $\lambda \sim 27$ cm, which by two times exceeds the length $L$. For this reason, at $p < 0.1$ Pa the fast argon atoms cannot appear due to charge-transfer collisions of argon ions in the space charge sheath surrounding the grid. In this case, the grid bombarded by 3-keV ions emits only two broad beams of 3-keV electrons propagating in opposite directions.

When the pressure grows to $p = 0.2$ Pa, accelerated from the plasma ions start to turn in the sheath into fast neutral atoms. The fast atoms leave the sheath and secondary ions appear therein. The energy of a fast atom is defined by the potential difference between the point of the charge-transfer collision and plasma. For this reason, the energy is distributed continuously from zero to ~$eU$. For secondary ions appearing due to charge transfer collisions, the energy is less than that of primary ions. A decrease in the ion energy diminishes the emission of electrons by the grid and current of electron beams.

At $p = 2$ Pa the charge-transfer length $\lambda \sim 1.3$ cm is close to the sheath width $d = 1.3$ cm. Most of the primary ions turn into fast atoms before reaching the grid and their maximal energy is less than $eU$. With the argon pressure increasing from 0.2 to 2 Pa, the energy of fast atoms is diminishing, their flow

density is increasing and the currents of electron beams are by an order of magnitude reduced. It is good for etching ceramic substrates and not suited for implantation of high-energy particles.

When the flat grid is replaced with a grid composed of parallel 50-cm-wide plates (Figure 2), the transparency grows to η = 0.94. However, it does not matter because, in this case, fast atoms do not appear in charge-transfer collisions, but due to contacts of accelerated ions with side surfaces of the grid plates. The advantage of the new grid is that fast atoms have the same energy, exactly equal to *eU*. This made it possible to form beams of neutral atoms with energy up to 30 keV in an effortless way, only applying high-voltage pulses to the grid hanged upon the feedthrough. Nevertheless, this method has serious drawbacks. A substantial part of electrical power is spent on the useless acceleration of secondary electrons emitted by the grid surface. Moreover, the chamber walls and products to be treated are coated with sputtered material of the grid.

Those drawbacks are eliminated, when the grid is placed on the brink of housing connected to the chamber. In this case, only edges of the grid plates are bombarded by accelerated ions. The low area of the grid surface, which emits secondary electrons, ensures a high-efficient use of electrical power. More than 90% of accelerated ions pass through the gaps between the grid plates and turn into fast atoms bombarding the substrates.

Due to the high uniformity of the plasma density distribution near the grid, the beam of high-energy helium or argon atoms is well collimated and ensures homogeneous treatment of the substrates. This is proven with experimental results of the etching the $Al_2O_3$ substrates by 3-keV argon atoms. During the etching at the discharge current of $I_d = 2$ A, the substrates were heated up to ~700 °C. It should be mentioned that the ion current density of $j_i = 10$ A/m$^2$ in the ion beam entering the grid substantially exceeds the mean value of the ion current density on the chamber walls $I_d/S = 2/1.5 = 1.3$ A/m$^2$, where $S = 1.5$ m$^2$ is the inner surface area of the chamber. Despite the homogeneous distribution of the plasma density in the central part of the chamber, the ion current density on the central part of the chamber door is by an order of the magnitude higher than in the corners between the doors and the cylindrical wall of the chamber.

Spectral diagnostic of the discharge plasma and beam of fast atoms showed that the atoms are monoenergetic and their energy is in good agreement with accelerating voltage applied to the grid. It was found that the irradiation of $Al_2O_3$ substrates with fast atoms of inert gases substantially decreases the friction. In the case of irradiation with 30-keV argon atoms for 90 s, the friction coefficient diminished from 0.37 to 0.15. Of course, the system for surface modification of dielectric substrates by broad beams of high-energy atoms based on the discharge with cold cathode can be used for irradiation of the substrates with fast atoms of any gases, including highly reactive gases.

**Author Contributions:** Conceptualization, A.M. and S.G.; formal analysis, S.G.; funding acquisition, S.G.; investigation, A.M., Y.M. and E.M.; methodology, M.V. and Y.M.; project administration, M.V.; resources, S.G. and M.V.; software, E.M.; supervision, S.G.; visualization, Y.M.; writing—original draft, M.V.; writing—review & editing, E.M. All authors have read and agreed to the published version of the manuscript.

**Funding:** This work is funded by the state assignment of the Ministry of Science and Higher Education of the Russian Federation, project No. 0707-2020-0025.

**Acknowledgments:** The work was carried out using the equipment of the Center of Collective Use MSUT "STANKIN".

**Conflicts of Interest:** The authors declare no conflicts of interest.

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
