# Peer review of "Surface Modification of Dielectric Substrates by Broad Beams of High-Energy Atoms of Inert Gases"

_technologies, doi:10.3390/technologies8030043_

Round 1
Reviewer 1 Report
The paper submitted by A.Metel et al. deals with the surface modification of dielectric substrates by high energy atoms generated in a hollow cathode type discharge.
The paper has too many problems as far as i can see, from the strong English language issues to a rather sluggish presentation of the results and omitting some important information with regard to their optical spectroscopy measurements.
Particularly in raw:
13, 25, 30 40-45, 50-57....
As well as a confusing introduction tot the discussion section where the authors present data from literature as opposed to tackle their own data. I also think that the discussions on the spectroscopic measurements cannot be taken face value without the error induced by the set up. Especially when they used the third decimal.
There are some interesting aspects to the paper. And if they work on in it can be of interested for the readers of the paper, however in this form i recommend it to be rejected.
Reviewer 2 Report
The submitted paper presents experimental investigations of the surface modification of dielectric substrates by fast atoms generated in different configurations of a negatively polarized grid (flat or with parallel plates) immersed or at the edge of a glow discharge.
The paper is generally well structured and clear, except in the transition from section 3 to 4. Throughout the text language needs improvement, especially the syntax of several sentences (order or words).
The content seems of interest, with good reference to previous works, and conclusions are well supported by experimental evidences.
Some local improvements are suggested in the detail comments:
Fig.1: Please add a reference dimension
Line 162-164 “For this reason, the beam of fast argon atoms bombarding the substrates is well collimated. This is confirmed by an abrupt transition between the substrate surfaces in Figure 3 … as well as a homogeneous distribution of the substrate etching rate” – Collimation is understood here as beam divergence, which is different from beam intensity uniformity. How a uniform width of the space charge sheath and a homogeneous etching rate distribution are related to a low divergence beam?
Line 200 “It can be concluded that fast atoms produced due to the charge neutralization …” – How is the neutralization demonstrated? The same spectrum could be produced by the ion beam experiencing charge exchange with the residual background gas.
Line 201 “… are practically monoenergetic” – There seem to be a significant fraction of energies in between the rest and the full energies: is it possible to show an intensity spectral profile? Normally these intermediate energies are explained as stripping losses in the accelerator: what in this case?
Line 221 “in our experiments, the irradiation time was equal to 30, 60, and 90 s.” – Please consider to show and compare the cases with different irradiation times in Figure 6
Line 241 – Discussion section starts with no link to the previous section and only at line 278 the link starts to be explained: some introduction at beginning of Discussion would benefit.
Line 266-271 – There is some confusion in the description of the behavior of ions and atoms, e.g. in “ions start to turn in the sheath into fast neutral atoms producing secondary ions and leaving the sheath.” once transformed into secondary ions do they still leave the sheath ?
Non exhaustive list of lines requiring language revision: 12-13, 35-36, 122-125, 175-177, 251-254,
Reviewer 3 Report
The manuscript under review is devoted to development a technology to treat dielectric surfaces with accelerated particles. Plasma discharge is utilized as the ion source and a grid of titanium plates is suggested as an ion beam neutralizer.
However, it is quite hard to read the manuscript due to very poor English. Too many questions appear. For example, I cannot understand which way authors measure the width of sheath between the plasma and the grid; how can author prove neutralization and so on.
So I strongly recommend improve English and resubmit the manuscript.
Reviewer 4 Report
The authors describe a method of neutral beam implantation for surface treatment to, for example, lower the friction coefficient of a material. Since the method relies on neutral beams it is suitable for dielectric substrates.
Inside a vacuum chamber a glow discharge of argon or helium ions is generated and at the plasma sheath, ions are accelerated. Depending on the pressure regime, these ions either keep their full energy or due to collisions in the sheath lose part or almost all of their energy. By using a grid structure, where the ions can neutralize during small angle scattering and using a certain pressure regime, they produce a beam of neutral, fast atoms for surface treatment. By placing the grid structure at the end of the plasma region they minimize wasted power due to the acceleration of electrons.
The authors use sputtering over a longer time frame to examine the uniformity of the beam, they utilize Doppler shifts to examine the velocity of the neutral atoms. Finally, they show results from a friction measurement of a treated vs untreated sample.
The main issue with the paper is that extensive language editing is unfortunately needed. In the current state, there are many places where the exact meaning is not clear or phrases are not used correctly (for example I'm not sure what the authors mean by "ion beams of chemical elements", I assume they mean atomic beams vs molecular beams?). Also the overall organization of the paper needs to be improved.
The authors do describe the experiment in detail, although in the current form, I find it hard to follow the description.
I believe that the authors need to improve their argument why the measured Doppler broadening will result in the energy of the implanted neutral ions and does not only measure the energy distribution of the charged ions across the sheath.
Although, I'm not an expert in friction measurement, I would like to see more than one data result from friction measurement to show that the experiment is actually repeatable. It seems that more samples where measured, in which case it would be good to show the measured data.
Finally the authors should discuss where they see limitation of this approach, e.g. in respect to achievable energy distribution and repetition rate and compare it to other methods to generate neutral beams.
For example, in the current setup ions are implanted at 30 keV. For argon in silicon this probably relates to an average depth of 25-30nm. The measured sputtering rate of the 3keV background plasma (when the 30kV pulse is not applied) was quoted at 600nm/h = 10nm/min. For the friction measurement a 1 minute implant was used. It seems that implants for longer than 3 minutes are already not possible or at least would limit the achievable dosee, since the implanted depth would be sputtered away.
For the submitted paper I therefore recommend acceptance after major revision and an additional review after editing for language has taken place.
I would encourage the authors to seek help from a native speaker or to use an editing service, so that their research can be easier to read and can reach a broader audience.
Round 2
Reviewer 1 Report
I appreciate that the authors reconsidered and resubmitted the paper, however I don't believe that they took my comments seriously, which is staggering. The changes that were made don’t raise enough the quality of the paper to grant an accept.
What I do not appreciate is the response of the authors, nor them being as defensive as they were. They should consider all the comments and respond according to the data presented in the paper, not discuss about the academic levels of their students (which I believe is out of the scope of this paper). I have to say I am offended by their lack of professionalism in their response.
The most important and surprising aspect is the fact that they still don’t discuss the resolution of the spectrometer, which is absurd. All the comments were made with a good intent and aimed to help improve the quality of the paper which in all fairness has some interesting results.
Sadly, I have to rejected the paper again.
Reviewer 3 Report
The experimental setup presented in the manuscript will operate with no doubts. It seems, the results obtained may be of interest to the reader. However, English requires much attention. I do believe author are able to find some person who can help them with deep proofreading.
Reviewer 4 Report
The authors addressed most concerns I had in regards to scientific question. I still believe that more friction measurement would be beneficial or the error needs to be discussed, e.g. it is not possible for me to understand what the typical error spread of these measurements are, e.g. are traces 2 and 3 in Figure 6 really different when taking the possible error into account or for that matter, are 1 and 2 separated by more than 1 sigma, 2 sigma or more?
My main concern is still the language. The author did minimal editing between the version and the text is still very hard to understand with many language mistakes. To list a few:
line 11: "plasma of glow discharge" -> "glow discharge plasma"
line 13: "beam of extracted from the" -> I assume the authors mean to say "beam of extracted ions", but it could also be beam of extracted neutral atoms?
line 15-16: During etching ... -> I do not understand this sentence at all. I assume the authors want to perhaps say "alumina substrates placed on the housing flange are heated to 700 C by the impinging fast neutral argon atoms", but I have to admit that I can only guess here
line 16: "Application to the grid of negative 30-kV pulses" -> "Application of negative 30-kV pulses to the grid"
...
line 30: "To generate the ion, accelerators of charged particles have been developed" -> missing "beams" after ion (this got changed incorrectly between versions), or it should be "to generate ions" (plural, no 'the')
line 37: "ions of chemical elements" -> I never heard this phrase and am not sure what they authors mean. I guess that the authors mean ion beams consisting of just one element? However, perhaps they refer to elemental beam vs molecular beams?
line 44: "To produce a beam of fast atoms it is needed first to generate..."-> "To produce a beam of fast atoms one needs to first generate"
Unfortunately, there are issues in almost every other line throughout the manuscript with many places where the meaning of the sentence is not clear. Furthermore, the text could be structured much better to say what the authors achieved and how they did it.
For example the abstract could be clearer. Here is a suggested abstract for the paper:
"""
We present a new method to generate a neutral beam for surface
treatment of materials by fast inert gases. The new method allows for
treatment at lower pressures enlarging the scope for glow discharge
applications. To generate the mono-energetic neutral beam a grid
composed of parallel plates is placed inside a vacuum chamber, a glow
discharge plasma is generated, and a beam is formed by
pulsing the grid to 30 kV to extract ions from the glow discharge. The
ions are then neutralized by small angle scattering at the surfaces of
the grid. By applying the high voltage for 50 us with a repetition
frequency of 50 Hz, heating of the target can be limited to 100 C
(instead of 700C when running continuous). We present results showing
the uniformity of the created beam and its energy distribution using
Doppler-shift measurement. Finally we show friction measurement of
treated metal pieces as a working example of an application of this
technology.
"""
Being not a native English speaker myself, I do understand that writing papers in another language is never easy. However, I do think that a clear presentation is necessary for the reader and in its current state the article is very hard to understand. I would suggest that the authors use an editing surface specialized on scientific/physics papers or work with a native speaker to improve the paper.
Round 3
Reviewer 1 Report
I feel that now the authors considered my observations and acted on it. Although i feel there is a dissonance with respect to one of my comments, the rest were considered so I recommend it for publication
Reviewer 3 Report
No doubts, authors have somehow improved the presentation of the results. I.e., the abstract have been deeply rearranged, which made it significantly readable and clear. On the other hand, despite some changes were made throughout the text, a lot of very strange sentences still present. To be clear, I show you a couple of examples.
1. line 36 "For the surface modification of engineering products is widely used the plasma immersion ion implantation" would be better to replace with
"To modify the surface of technical parts, plasma immersion ion implantation is widely used"
2. line 47 "Then one needs to convert the ions into fast neutral atoms due to charge exchange collisions with gas molecules and separate them from residual ions using a magnetic field" to replace with
"Then it is necessary to convert ions into fast neutral atoms by charge exchange collisions with gas molecules and to separate atoms from residual ions using a magnetic field."
Same you write in any 10 lines. I do believe this is not a referee's task to make this corrections. So find somebody else more fluent in English.
Reviewer 4 Report
The main issue is still the language. The authors did minor changes, but the paper is still hard to read. There are still too many language issues to list them all here. Here are two examples of sentences in the first two pages that need to be changed before publication:
line 36: "For the surface modification of engineering products is widely used the plasma immersion ion implantation [15]."
This is not a valid English sentence. The authors probably want to say "Plasma immersion ion implantation is a widely used surface modification technique applied to a wide range of products."
line 61: [The goal is to develop a technology which could] "enlarge the surface area of the substrates."
I'm pretty sure the authors are not planning to increase the actual surface area of the substrates, but to develop a technology that can treat substrates with larger surface areas?
